# Cohesins and condensins orchestrate the 4D dynamics of yeast chromosomes during the cell cycle

Luciana Lazar-Stefanita[1,2,3,4], Vittore F Scolari[1,2,3], Guillaume Mercy[1,2,3,4], Héloise Muller[1,2,3], Thomas M Guérin[5], Agnès Thierry[1,2,3], Julien Mozziconacci[6,7,*] ![ID] & Romain Koszul[1,2,3,**] ![ID]

## Abstract

Duplication and segregation of chromosomes involves dynamic reorganization of their internal structure by conserved architectural proteins, including the structural maintenance of chromosomes (SMC) complexes cohesin and condensin. Despite active investigation of the roles of these factors, a genome-wide view of dynamic chromosome architecture at both small and large scale during cell division is still missing. Here, we report the first comprehensive 4D analysis of the higher-order organization of the *Saccharomyces cerevisiae* genome throughout the cell cycle and investigate the roles of SMC complexes in controlling structural transitions. During replication, cohesion establishment promotes numerous long-range intra-chromosomal contacts and correlates with the individualization of chromosomes, which culminates at metaphase. In anaphase, mitotic chromosomes are abruptly reorganized depending on mechanical forces exerted by the mitotic spindle. Formation of a condensin-dependent loop bridging the centromere cluster with the rDNA loci suggests that condensin-mediated forces may also directly facilitate segregation. This work therefore comprehensively recapitulates cell cycle-dependent chromosome dynamics in a unicellular eukaryote, but also unveils new features of chromosome structural reorganization during highly conserved stages of cell division.

**Keywords** chromosome segregation; Hi-C; loop extrusion; replication profile; SMC
**Subject Categories** Cell Cycle; Chromatin, Epigenetics, Genomics & Functional Genomics; DNA Replication, Repair & Recombination
**The EMBO Journal (2017) 36: 2684–2697**

See also: **C Barrington *et al*** (September 2017)

## Introduction

The chromosomes of prokaryotes and eukaryotes display multiple levels of hierarchical organization, whose dynamic changes influence or regulate metabolic processes including gene expression and DNA replication and repair (Taddei & Gasser, 2012; Wang *et al*, 2013; Dekker & Mirny, 2016). The improper coordination of chromosome condensation and segregation during the cell cycle can lead to important structural abnormalities and result in cell death or diseases such as cancer (Valton & Dekker, 2016). In recent years, major advances in imaging and chromosome conformation capture approaches (Dekker *et al*, 2002; Lieberman-Aiden *et al*, 2009; 3C, Hi-C) have complemented earlier work by describing at an unprecedented resolution the multiple hierarchical layers of genome organization. A variety of remarkable 3D chromosomal structures have been described in a number of species, including in unicellular organisms such as bacteria and yeasts.

The genome of budding yeast *Saccharomyces cerevisiae* presents a Rabl organization driven by (i) centromeres clustering at the spindle pole body (SPB, *S. cerevisiae* microtubule organizing center), (ii) telomeres tethering to the nuclear envelope, (iii) the nucleolus where the rDNA is sequestered opposite to the SPB, and (iv) chromosome arm length (Burgess & Kleckner, 1999; Taddei & Gasser, 2012). Hi-C experiments have confirmed this Rabl organization, but the existence of sub-megabase structures within yeast chromosomes similar to mammalian topological associated domains or their bacterial equivalent is still controversial (Duan *et al*, 2010; Hsieh *et al*, 2015; Eser *et al*, 2017). Importantly, genomic analysis of chromosome 3D architectures has usually been done using asynchronous populations, in which cells are found in various stages of the cell cycle. However, the initiation and progression of replication, followed by the segregation of the sister chromatids (SCs) into daughter cells, is expected to modify the genome higher-order

1 Institut Pasteur, Department Genomes and Genetics, Unité Régulation Spatiale des Génomes, Paris, France
2 CNRS, UMR 3525, Paris, France
3 Institut Pasteur, CNRS, Center of Bioinformatics, Biostatistics and Integrative Biology (C3BI), USR 3756, Paris, France
4 Sorbonne Universités, UPMC Université Paris 6, Complexité du Vivant, Paris, France
5 Laboratoire Télomères et Réparation du Chromosome, CEA, INSERM, UMR 967, IRCM, Université Paris-Saclay, Fontenay-aux-Roses, France
6 Sorbonne Universités, Theoretical Physics for Condensed Matter Lab, UPMC Université Paris 06, Paris, France
7 CNRS, UMR 7600, Paris, France
*Corresponding author. Tel: +33 1 44 27 45 40; E-mail: mozziconacci@lptmc.jussieu.fr
**Corresponding author. Tel: +33 1 40 61 33 25; E-mail: romain.koszul@pasteur.fr

organization. Recent studies have unveiled cell-cycle stage-specific genome-wide topological variations in bacteria, yeast, fly, and mammals (Naumova *et al*, 2013; Guidi *et al*, 2015; Marbouty *et al*, 2015; Hug *et al*, 2017). As expected, in all species the largest reorganization transition is associated with SC condensation, a fundamental process occurring concomitantly to their individualization, and facilitating their proper segregation.

Pioneer studies on yeasts proved essential to study these processes. Mutations in cell-division cycle (*cdc*; Hartwell *et al*, 1973) genes can block the cell cycle progression, enabling the study of global and/or local chromosome reorganization at specific cycle phases (Hartwell *et al*, 1973; Guacci *et al*, 1994; Sullivan *et al*, 2004; Renshaw *et al*, 2010; Rock & Amon, 2011). The evolutionary conserved structural maintenance of chromosomes (SMC) proteins bind to chromosomes and modify their structure in spatially and temporally regulated manner during the cell cycle (Aragon *et al*, 2013; Uhlmann, 2016). Cohesins, such as Scc1, promote SC cohesion during DNA replication (Blat & Kleckner, 1999; Glynn *et al*, 2004) and get cleaved at the metaphase-to-anaphase transition (Uhlmann *et al*, 1999). At the same time, condensins such as Smc2 are loaded onto SCs to facilitate their segregation (Renshaw *et al*, 2010; Stephens *et al*, 2011; Hirano, 2012). In fission yeast, the binding of SMCs modifies the level of chromosome compaction defined as the ratio between long (> 10 kb)- and short-range (< 10 kb) contacts at specific loci (Mizuguchi *et al*, 2014; Kim *et al*, 2016).

While Hi-C studies on mammalian and *drosophila* cells have confirmed this compaction change and provided important insights on the organization of mitotic chromosomes' internal structure (Naumova *et al*, 2013; Hug *et al*, 2017), no comprehensive analysis of the 4D dynamics of the chromosomes during an entire eukaryotic cell cycle has been achieved. To explore new chromosomal structural features over the cell cycle progression, we analyzed the internal folding and overall organization of *S. cerevisiae* genome over 15 synchronized time points and the role of cohesin and condensin using Hi-C (Dekker *et al*, 2002; Lieberman-Aiden *et al*, 2009). This analysis provides a broad overview and in-depth insight on SMC-dependent structural transitions resulting in chromosome individualization and segregation, including a potential role for a condensin-dependent loop in contributing to the segregation of the rDNA cluster.

## Results

### Comparison of chromosome contact maps of synchronized cells

Hi-C libraries were generated from cell cultures synchronized in G1 with elutriation (Marbouty *et al*, 2014) and/or arrested at different stages of the cell cycle through thermosensitive (ts) *cdc* mutations (Fig 1A; Hartwell *et al*, 1973). After sequencing, the corresponding normalized genome-wide contact maps were computed (bin: 5 kb; Fig 1C and D, left panels; Fig EV1; Materials and Methods; Cournac *et al*, 2012).

These 2D maps were translated into 3D representations to visualize the main folding features (Lesne *et al*, 2014; e.g., centromeres

and telomeres clustering in G1; Figs 1B and EV1). These 3D structures are average representations of the contact frequencies quantified over a population of cells and therefore do not represent the exact structure found in individual cells. For instance, on these 3D representations all the telomeres loosely cluster together. In a single nucleus, telomeres rather form small groups scattered all around the nuclear membrane (Taddei & Gasser, 2012). Since in different cells the composition of these clusters differs, all telomeres end up being regrouped together in the average 3D structure that reflects the population average of contacts. In addition, they are not polymer models and cannot be interpreted as such. Nevertheless, these representations conveniently highlight important structural features not readily apparent in the 2D maps (Mercy *et al*, 2017).

The differences between two conditions were determined by computing the log-ratio between the maps (bin: 50 kb[†]; Fig 1C; Materials and Methods). The color scale reflects the variations in contact frequency for each bin between two different contact maps. The ratio of contact maps generated from two independent G1 cell populations (experimental replicates) displays a relatively homogenous white (i.e., null) signal, corresponding to little differences between them (Fig 1C, right panel). These minor variations between the maps result in occasional faint colored areas and reflect experimental noise (Appendix and Materials and Methods). On the other hand, the ratio between exponentially growing G1 and quiescent G0 cells contact maps (Fig 1D, right panel) shows a strong difference in inter-telomere contact frequencies, reflecting the formation of the telomeres hyper-cluster characteristic of the G0 metabolic state (Guidi *et al*, 2015; Fig 1D, black arrowheads).

Multiple maps can also be compared altogether by computing their pairwise distance matrix, showing that the genome organization of cells in anaphase (*cdc15*) differs the most compared to other time points (Fig 1E; Materials and Methods). The overall similarities/differences between datasets can then be summarized using principal component analysis (PCA; Fig 1F). This 2D representation shows that the experimental duplicates (such as G1, or anaphase *cdc15*) clustered together, while the distance increases progressively between G1 (obtained with either elutriation or *cdc6* ts mutant), metaphase (*cdc20*), and the distant anaphase (*cdc15*) datasets.

Altogether, these comparisons highlight major changes in chromosome higher-order architecture taking place in cells progressing throughout the cell cycle into metaphase and anaphase.

### Cohesin-mediated compaction during S phase

To decipher the chromosome structural changes that take place during replication, synchronized G1 cells were released into S phase and Hi-C maps generated for six time points sampled from two independent kinetics (Figs 2A and EV2; Materials and Methods). The PCA reveals a progressive structural evolution from G1 to late S/G2 phase (Fig 2B). The dependency of the contact probability *P* on genomic distance reflects the chromosome compaction state (Lieberman-Aiden *et al*, 2009; Naumova *et al*, 2013; Mizuguchi *et al*, 2014). The *P(s)* shows a gradual and consistent enrichment in long-range intra-chromosomal contacts (> 20 kb) with respect to

---

[†]Correction added on 15 September 2017 after first online publication: Bin size was corrected from 5 to 50 kb.

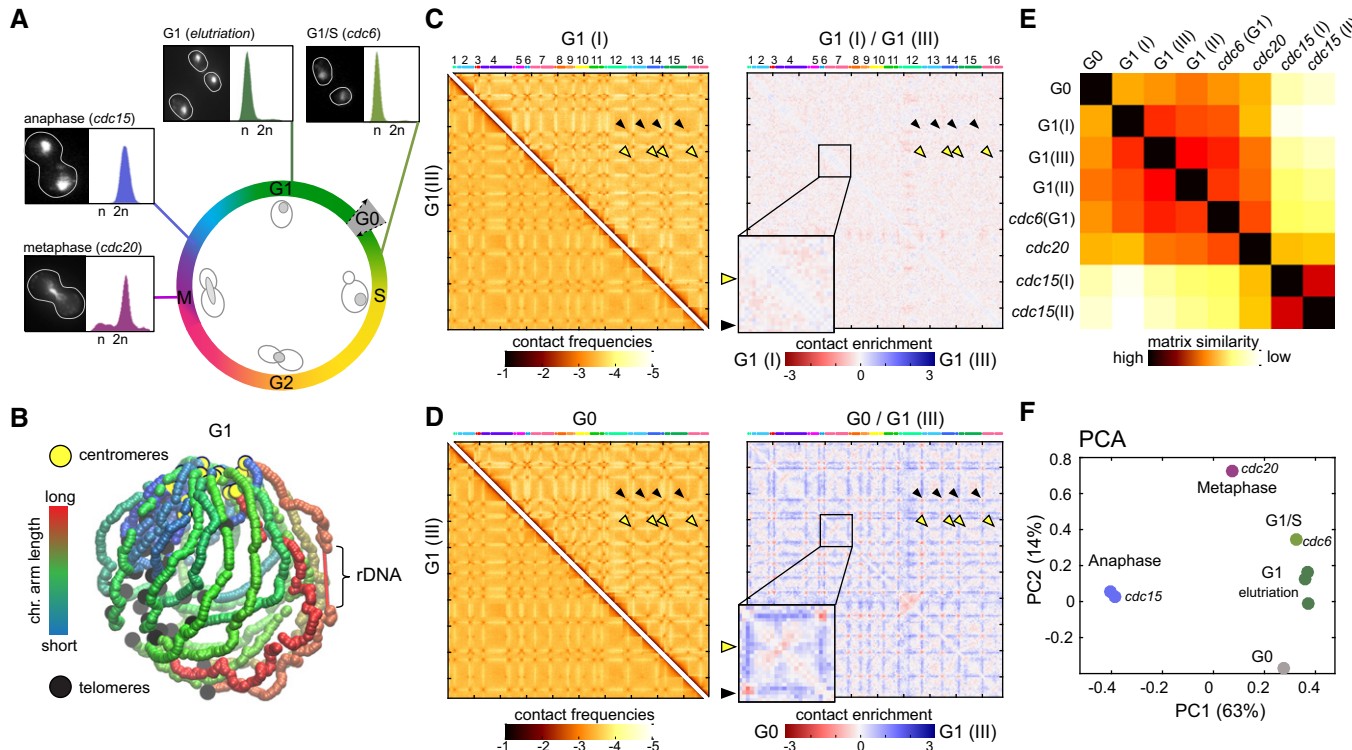

**Figure 1. Comparison of genome structures recovered from five synchronized stages over the cell cycle.**

A Overview of the different synchronization time points with corresponding FACS profiles and representative images of DAPI-stained cells.

B 3D average representation of the Hi-C contact map of a yeast G1 population. The color code reflects chromosomal arm lengths, and centromeres, telomeres, and rDNA are highlighted.

C, D Comparison of contact maps. The 16 yeast chromosomes are displayed atop the maps. Black arrowheads: inter-telomere contacts. Yellow arrowheads: inter-centromeric contacts. Left panels: Hi-C maps obtained from two G1 cell populations synchronized independently (C) and from G1 and G0 populations (D). Brown to yellow color scales reflect high to low contact frequencies, respectively (log10). Right panels: log-ratio between each pair of maps. Insets display magnifications of chr4. Blue to red color scales reflect the enrichment in contacts in one population with respect to the other (log2).

E Pairwise Euclidian distances between contact maps of populations of G0, G1 either synchronized with elutriation or blocked using a *cdc6* mutant, metaphase (*cdc20* mutant), and anaphase (*cdc15* mutant) cells. Color code: contact map similarity.

F Principal component analysis (PCA) of the distance matrix in (E).

short-range (< 10 kb) during replication (Fig 2C). This compaction change is absent when replication is impaired, for instance, in the absence of the replication-checkpoint regulator *cdc6* (Piatti *et al*, 1995), even though cells enter mitosis and engage into segregation of non-replicated chromosomes (Fig 2D, left panel). The progressive increase in long-range contacts stops with the completion of S phase, when it reaches the level observed in cells arrested at the G2/metaphase transition (G/M) with the microtubule-depolymerizing drug nocodazole (Jacobs *et al*, 1988; Fig 2D, middle panel). The crossing of the *P(s)* slopes from the early to late replication time points occurs around 10–20 kb (Fig 2C, highlighted in gray), a window within the range of the spacing reported between cohesin binding sites (~11 kb on average; Glynn *et al*, 2004), suggesting that this change in compaction could be due to cohesin activity. In agreement with the key role of cohesin in sister-chromatid folding during replication, Scc1 depletion using an auxin-inducible degron *scc1-aid* strain prevents the enrichment in long-range contacts in S/G2 (Fig 2D, right panel). This result supports the hypothesis that distant regions enriched in cohesin are tethered together, resulting in chromatin loops (Guillou *et al*, 2010).

## Chromosome compaction is concomitant with chromosome individualization

The Scc1-dependent compaction occurs concomitantly with a gradual individualization of the SC pairs throughout replication, as shown by the overall increase in the ratio between intra- and inter-chromosomal contacts from 63 ± 10% in G1 (six time points) to 73 ± 4% in S/G2 (four time points) and illustrated by the ratio between G1 and G2 maps (Fig 2E, top right ratio). In sharp contrast to this overall decrease in inter-chromosomal contacts, the centromeres of different chromosome tend to strongly cluster in G2. In the absence of the cohesin Scc1, intra-chromosomal contacts in G2 cells decrease to levels similar to or even below G1 (Fig 2E, bottom left ratio), while the major binding sites for cohesin (i.e., centromeres; Glynn *et al*, 2004) also exhibit a reduced level of contacts (Fig 2F; Appendix Fig S1). These results suggest that cohesins affect the genome organization through the gradual compaction of SC, the clustering of centromeres, and chromosome individualization. Although yeast chromosomes are shorter than mammalian chromosomes, they similarly change their internal

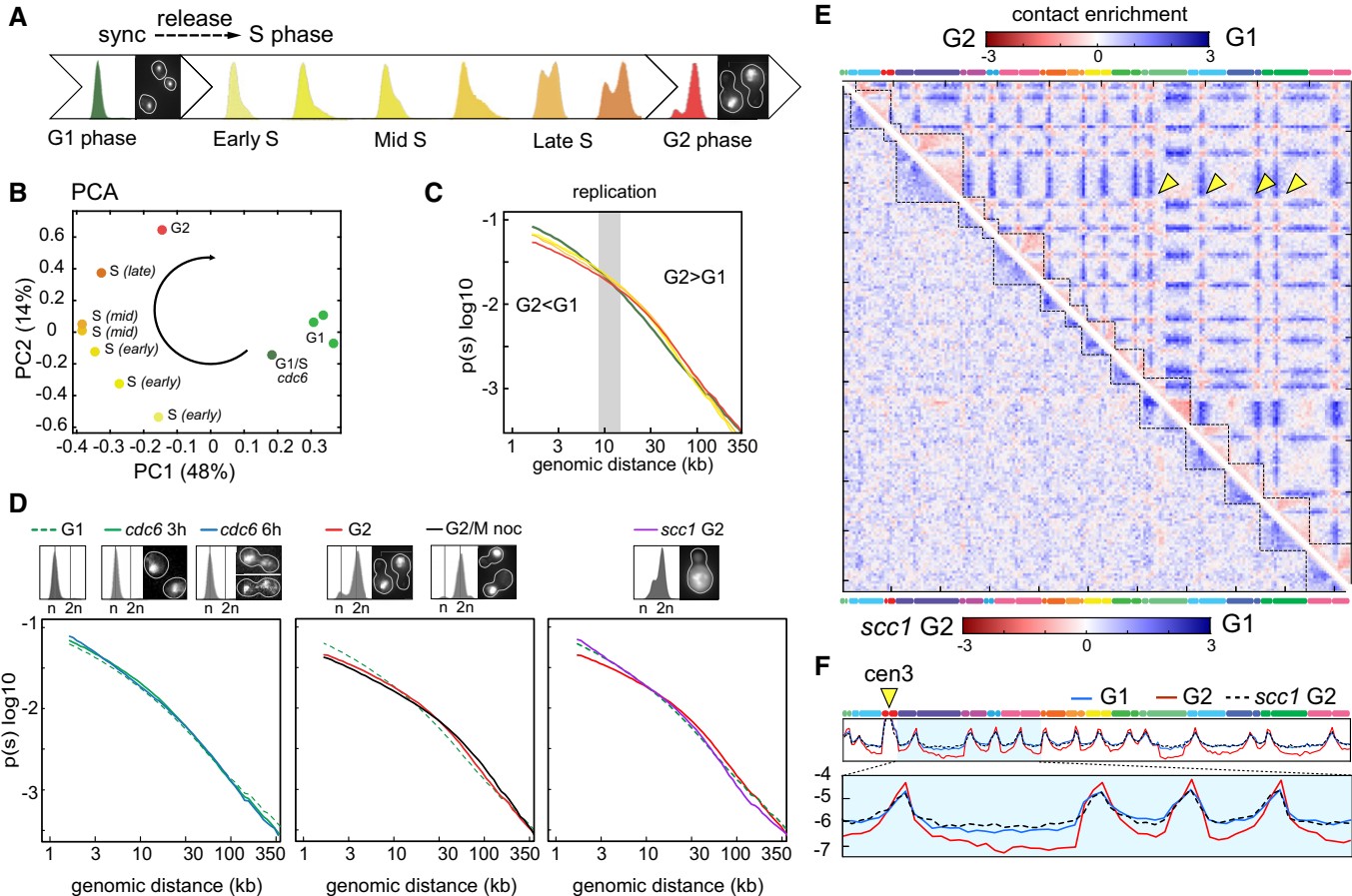

**Figure 2. Dynamic reorganization of chromosomes during replication.**

A   FACS profiles and representative DAPI-stained cells of G1 synchronized cells released in S phase.

B   PCA of the distance matrix between the contact maps of the population displayed in (A).

C   *P*(*s*), that is, average intra-chromosomal contact frequency *P* between two loci with respect to their genomic distance *s* along the chromosome (log–log scale) during replication (color code identical to FACS profiles and PCA).

D   Left panel: *P*(*s*) of replication-defective cells (*cdc6* thermosensitive mutant). G1 elutriated cells were released for 3 h and 6 h in non-permissive conditions. The corresponding FACS profiles show no S-phase progression. Middle panel: *P*(*s*) of cells that completed replication. G1 elutriated cells were released in S phase in the absence or presence of nocodazole (G2/M noc). Right panel: *P*(*s*) of cohesin-depleted (*scc1* G2) and nocodazole-arrested cells.

E   Log-ratio of contact maps between G2 and G1 cells (top right) and *scc1* G2 and G1 cells (bottom left). Blue to red color scales reflect the enrichment in contacts in one population with respect to the other (log2). Yellow arrowheads: inter-centromere contacts.

F   Normalized contact frequencies between chr3 centromere (cen3; yellow arrowhead) and the rest of the genome for G1, G2, and *scc1* G2.

conformation and individualize themselves prior to entering metaphase.

**Spatial resolution of the replication timing program**

In budding yeast, replication initiates at discrete autonomously replicating sequences (ARSs; Brewer & Fangman, 1987). ARSs display partially stochastic activation, with only a subset of origins activated early during S phase. The distribution of early origins is uneven, with an enrichment in pericentromeric regions, and a depletion in subtelomeric regions. The genome-wide pattern of ARS activation timing defines a population-average replication timing program (Raghuraman *et al*, 2001). To investigate the link between genome organization and replication timing, the read coverage of the Hi-C libraries was used to compute the replication timing profile of the cell population for each of the time point, and follow their progression

through S phase. The average profile correlates well with previously published pattern (Raghuraman *et al*, 2001; McCune *et al*, 2008; Fig 3A; Materials and Methods). To visualize the progression of replication on the higher-order architecture of the genome, we colored the 3D structures recovered from three early replication time points according to their replication progression status. The superimposition of the three structures recapitulates intuitive properties of yeast replication program, with a "replication wave" propagating from the centromeric regions enriched in early origins, through chromosomal arms, and toward the late replicating subtelomeric regions (Fig 3B and C; red and blue signal, respectively).

We also asked whether our data support the proposed co-localization of adjacent early replication origins (Kitamura *et al*, 2006; Knott *et al*, 2012; Saner *et al*, 2013). We found a statistically significant enrichment in contacts between these positions and their surrounding regions, but whether it results from an active

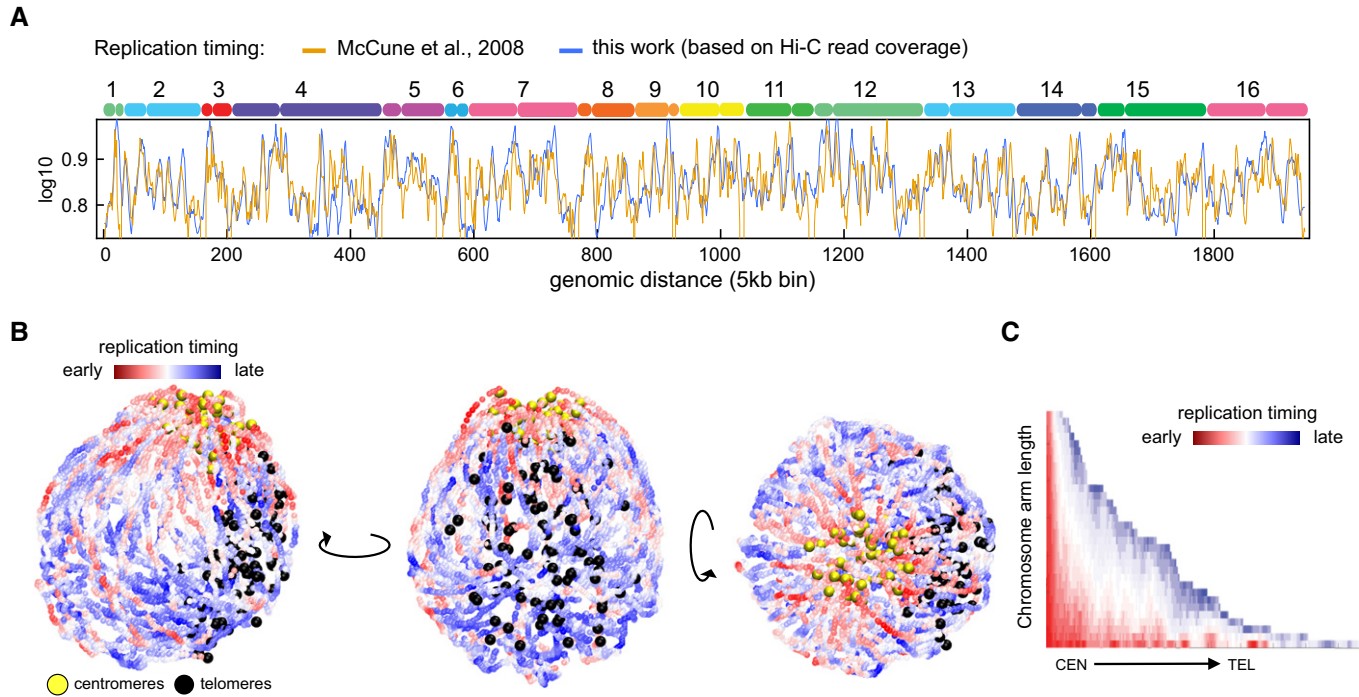

**Figure 3. 3D replication profile.**

A  Comparison of replication profiles of the synchronized populations used for the analysis displayed in Fig 2G. The read coverage of raw Hi–C libraries reflects the replication progression throughout S phase, plotted along the 16 chromosomes of the yeast genome (top axis; blue curve). The replication timing obtained in this study is highly similar to the one from McCune *et al* (2008) (yellow curve).

B  Superposition of three 3D representations of chromosomes in early replication (I, II, III). The color scale indicates the replication timing. Centromeres and telomeres are highlighted. Different views of the structure are presented.

C  Pattern of the replication profile for each of the chromosomal arms. The color code reflects the timing of replication.

co-localization or from their positioning in the pericentromeric regions co-localized due to the Rabl organization remains unclear (not shown). More analyses are required to solve this question and integrate the different observations.

## Global structural changes during mitotic transitions

After replication, cells progress into mitosis (M phase). During metaphase, microtubules originating from opposite SPBs attach to the kinetochores of the two SCs (London & Biggins, 2014). The anaphase-promoting complex (APC) co-activator Cdc20 is essential for the proper activation of separase, resulting in the cleavage of cohesin and SC segregation in anaphase (Uhlmann *et al*, 1999; Visintin *et al*, 1997; 20). In the absence of Cdc20, cohesins are not cleaved and cells remain blocked in metaphase. Another key player in mitosis progression is the Cdc15 kinase which promotes mitotic exit at the end of anaphase by activating cytokinesis (Rock & Amon, 2011). In the absence of Cdc15, cells are therefore blocked into late anaphase. The higher-order changes in the organization of chromosomes that take place during metaphase and anaphase were investigated using populations of cells synchronized with conditional mutants of *cdc20* and *cdc15*, respectively. Contact maps of *cdc20*-, *cdc15*-, and *cdc15*-arrested cells released into permissive conditions were generated to characterize chromosome reorganization throughout M phase (Figs 4A and EV3; Materials and Methods). PCA shows that the major structural change occurs during mitotic exit and that

cells released from the *cdc15* arrest display after 60 min a G1-like genome structure, reflecting the fact that the entire cell cycle is now covered by our analysis (Fig 4B). The $P(s)$ reveals a strong increase in short-range contacts (< 10–20 kb) from G2 to anaphase, exceeding G1 levels which are only restored after anaphase completion (Fig 4C, left panel). This increase in short-range contacts and the accompanying drop in long-range contacts suggest the formation of an elongated, stretched structure. Upon spindle destabilization using the microtubule-depolymerizing drug nocodazole in *cdc15*-arrested cells (*cdc15* noc), the two segregated chromosomal masses get closer as shown by imaging of DAPI-stained cells (Fig 4C, inset; Fig EV4; Materials and Methods), in agreement with former reports (Jacobs *et al*, 1988). In these cells, the stretched chromosomal structure disappears as shown by a $P(s)$ that now overlaps the G2 curve (Fig 4C, right panel). Besides the change in $P(s)$, the global contact pattern of *cdc15*-arrested cells remains unaltered following nocodazole treatment (Fig 4D, upper right ratio). Altogether, these results show that microtubule-dependent segregation forces contribute to the stretching the chromosomes in anaphase, possibly in combination with additional constraints resisting this force such as the cohesion of SC arm extremities (see Discussion).

## Nocodazole affects chromosome 12 conformation

Nocodazole is commonly used to synchronize cells at the G2/M transition. We took advantage of having contact maps of

*cdc20*-arrested cells in metaphase to compare them with those obtained from nocodazole-arrested cells (Fig EV4; Materials and Methods). The ratio map appeared globally similar, although we noticed in the presence of nocodazole a small drop in inter-chromosomal contacts (Fig 4D, bottom left ratio). Chromosome 12 (chr12)

also presents a peculiar signal at the level of the rDNA cluster (Fig 4E, left panel), with an enrichment in contacts between the two flanking regions of the rDNA cluster in G2/M nocodazole-treated cells compared to *cdc20*-arrested cells (Fig 4E, right panel). These results indicate that the G2/M nocodazole arrest is associated with a

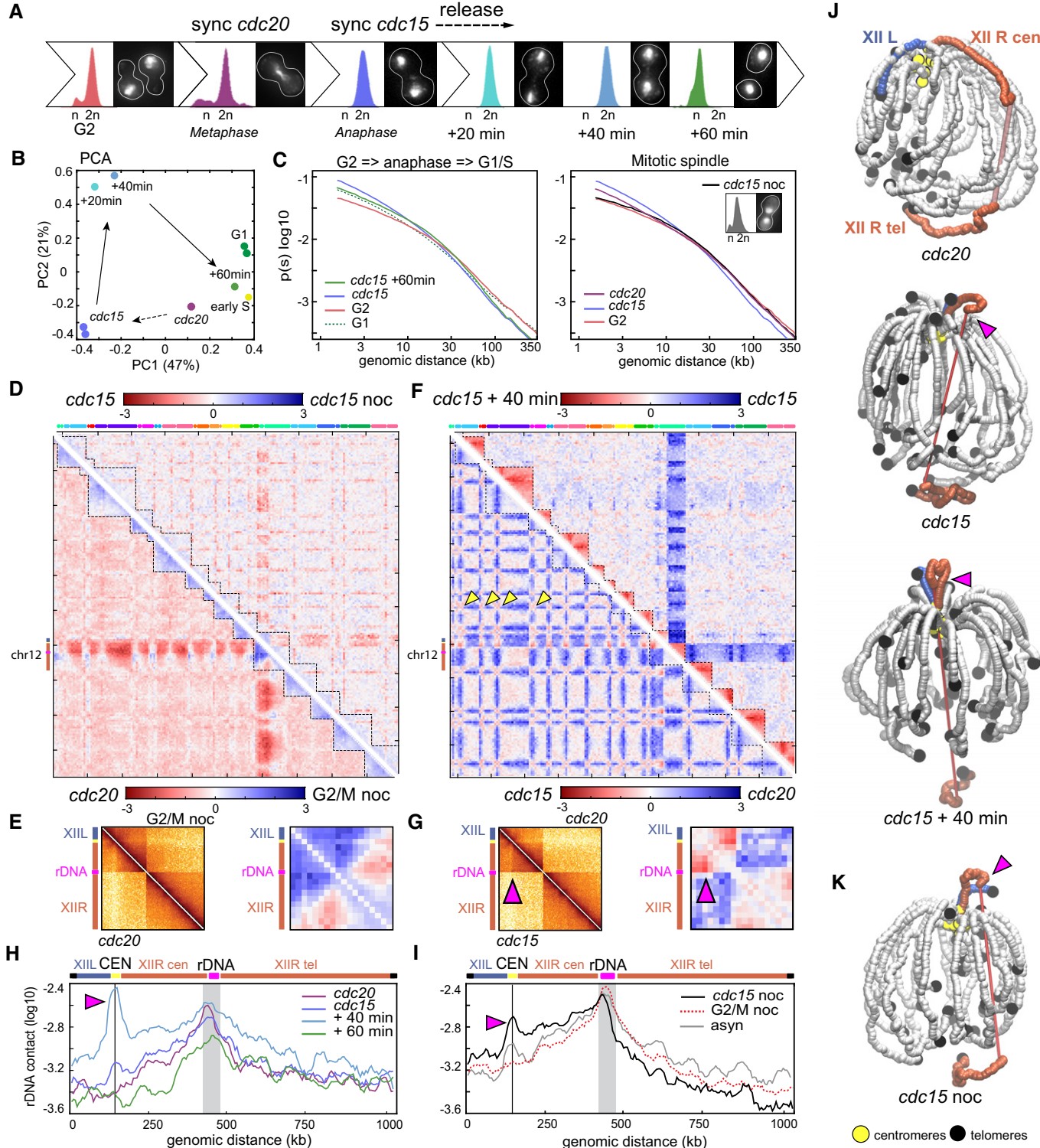

**Figure 4.**

destabilization of the chr12 structure at the level of the rDNA locus. The intra-chromosomal contact increase within chr12 is also accompanied by a global decrease in inter-chromosomal contacts in the presence of nocodazole (Fig 4D, bottom left ratio). Remarkably, chr12 organization was not affected when *cdc15*-arrested cells were treated with nocodazole (Fig 4D, top right ratio). Altogether, these observations point to a role for the microtubule array in maintaining the organization of the nucleolus inside the nucleus, before its segregation in anaphase. In summary, while chromosome structures are overall similar in cell synchronized in G2 by nocodazole or in a *cdc20* ts mutant, nocodazole-arrested cells present a slightly different nucleolus structure (and, by extension, chr12). One interpretation could be that the condensation of the rDNA is not yet completed in G2/M nocodazole arrest and that as a result, rDNA flanking regions are freer to contact each other's.

## Chromosome 12 looping during anaphase

The comparison of *cdc15* and *cdc20* maps shows an increase in centromere clustering in anaphase, leading to the formation of a prominent polymer brush structure (Daoud & Cotton, 1982; Fig 4F, bottom left ratio, yellow arrowheads). Such increase is in agreement with the role of condensin in forming a "spring" of chromatin at pericentromeric regions at the metaphase-to-anaphase transition (Stephens *et al*, 2011). Surprisingly, a peculiar loop pattern appears on chr12 in *cdc15*-arrested cells, bridging the centromere and the centromere–proximal left flanking region of the rDNA cluster (see pink arrowheads in Fig 4F, G and H). Upon release from the *cdc15* arrest, the telomere–proximal right flanking region of the rDNA cluster becomes strongly isolated from the rest of the genome (Fig 4F, upper right ratio; *cdc15*+40 min), while the contacts of the centromere–rDNA loop intensify (Fig 4H; *cdc15*+40 min). After completion of mitosis and re-entry in interphase (*cdc15*+60 min), the loop disappears (Fig 4H). Interestingly, this loop can be seen in asynchronous populations while it is only present in anaphase (Fig 4I). 3D representations illustrate the dramatic reorganization of chr12 and the formation of the loop bridging centromeric region and the rDNA (Fig 4J, pink arrowheads). Microtubules are not required to maintain this loop in anaphase, since it remains present in

*cdc15*-arrested cells treated with nocodazole (Fig 4D, upper right ratio; Fig 4I and K), suggesting that the left flanking region of the rDNA is physically bound through an unknown mechanism to the centromeric regions. These results complement imaging studies showing that the rDNA exhibits a dense, line-like shape that extends throughout the nucleus at anaphase (2.1 SD, 0.2 μm; Sullivan *et al*, 2004).

## Condensin promotes dramatic reorganization of chromosomes during anaphase

The proper condensation and segregation of the rDNA cluster requires the nucleolar release of the Cdc14 phosphatase. Cdc14 mediates a shutdown of rDNA transcription, facilitating the loading of the Smc2 condensin and hence the condensation of the cluster (Yoshida *et al*, 2002; D'Amours *et al*, 2004; Sullivan *et al*, 2004, 14; Machín *et al*, 2006; Clemente-Blanco *et al*, 2009). In addition, topoisomerase II (Top2), which decatenates the intertwining structures that appear between SCs during replication, is also required for rDNA segregation to proceed (Sullivan *et al*, 2004; D'Ambrosio *et al*, 2008; Baxter *et al*, 2011; Leonard *et al*, 2015). We investigated the influence of those factors on the 3D structure of the rDNA locus during anaphase (Figs 5A and EV5; Materials and Methods).

First, Smc2 depletion in *smc2-aid cdc15*-arrested strain affects anaphase genome organization by (i) reducing centromere clustering and (ii) suppressing the formation of the rDNA loop, with a resulting contact map highly similar to the *cdc20* map (Fig 5B, bottom left ratio). Therefore, condensins are responsible for the observed increase in inter-centromere contacts at anaphase compared to metaphase (Fig 4F, bottom left ratio), while they are also required for the formation of the loop bridging the centromere of chromosome 12 with the rDNA cluster (two loci enriched in condensin deposition). The *smc2 cdc15* and *cdc14* maps are strikingly similar (Fig 5C, bottom left ratio). The 3D representations of *smc2 cdc15* and *cdc14* cells (Fig 5E) and the rDNA contact plots with the rest of chr12 (Fig 5F) illustrate the loss of the rDNA loop in the absence of Smc2 and/or Cdc14. In addition to this effect, both mutants also display the same decrease in centromere clustering compared to *cdc15* cells (Fig 5C, upper right ratio; Fig 5G), pointing at their functional relationship on the same pathway.

◀ **Figure 4. Dynamic reorganization of chromosomes during mitosis.**

A    FACS profiles and representative DAPI-stained cells of synchronized and/or released populations, from G2 until re-entry in G1/S.

B    PCA of the distance matrix between the contact maps of the populations described in (A).

C    Left panel: *P(s)* of cells in G1, G2, and anaphase (*cdc15*) and released from a *cdc15* arrest (*cdc15*+60 min). Right panel: *P(s)* of G2, *cdc20*-, and *cdc15*-arrested cells in the absence or presence of nocodazole (*cdc15* noc).

D    Log-ratio of contact maps. Bottom left: ratio between cells arrested in metaphase (*cdc20*) or at the G2/M transition with nocodazole (G2/M noc). Top right: ratio of cells blocked in anaphase and treated or not with nocodazole (*cdc15* noc and *cdc15*, respectively). Blue to red color scales reflect the enrichment in contacts in one population with respect to the other (log2).

E    Left: chr12-normalized contact maps of cells arrested at the G2/M transition and *cdc20*-arrested cells. Right: magnification of the log-ratio map from (D, bottom left).

F    Log-ratio of contact maps. Bottom left: log-ratio between *cdc20*- and *cdc15*-arrested cells. Top right: log-ratio of *cdc15*-arrested and *cdc15*-released (40 min) cells. Blue to red color scales reflect the enrichment in contacts in one population with respect to the other (log2). Yellow arrowheads: inter-centromere contacts.

G    Left: chr12-normalized contact maps in *cdc20*- and *cdc15*-arrested cells. Right: magnification of the log-ratio map from (F, bottom left). Pink arrowheads point at the right arm anaphase loop.

H, I    Distributions of intra-chromosomal contacts made by a 20-kb cen-proximal rDNA flanking region (highlighted in gray) with the rest of chr12 in *cdc20*-, *cdc15*-, *cdc15*-released (+40 min, +60 min), nocodazole-treated (G2/M noc, *cdc15* noc), and asynchronous (asyn) cells. Schematic representations of chr12 are displayed atop the graphs. Gray lines indicate centromere position. Pink arrowheads point at the right arm anaphase loop.

J    3D representations of the contact maps from *cdc20*- and *cdc15*-arrested and *cdc20*- and *cdc15*-released (+40 min) cells. The right (XIIR) and left (XIIL) arms of chr12 are highlighted in red and blue, respectively. Pink arrowheads point at the right arm anaphase loop. Centromeres and telomeres are highlighted.

K    3D representation of the contact map from *cdc15* noc cells. Pink arrowhead points at the right arm anaphase loop.

     

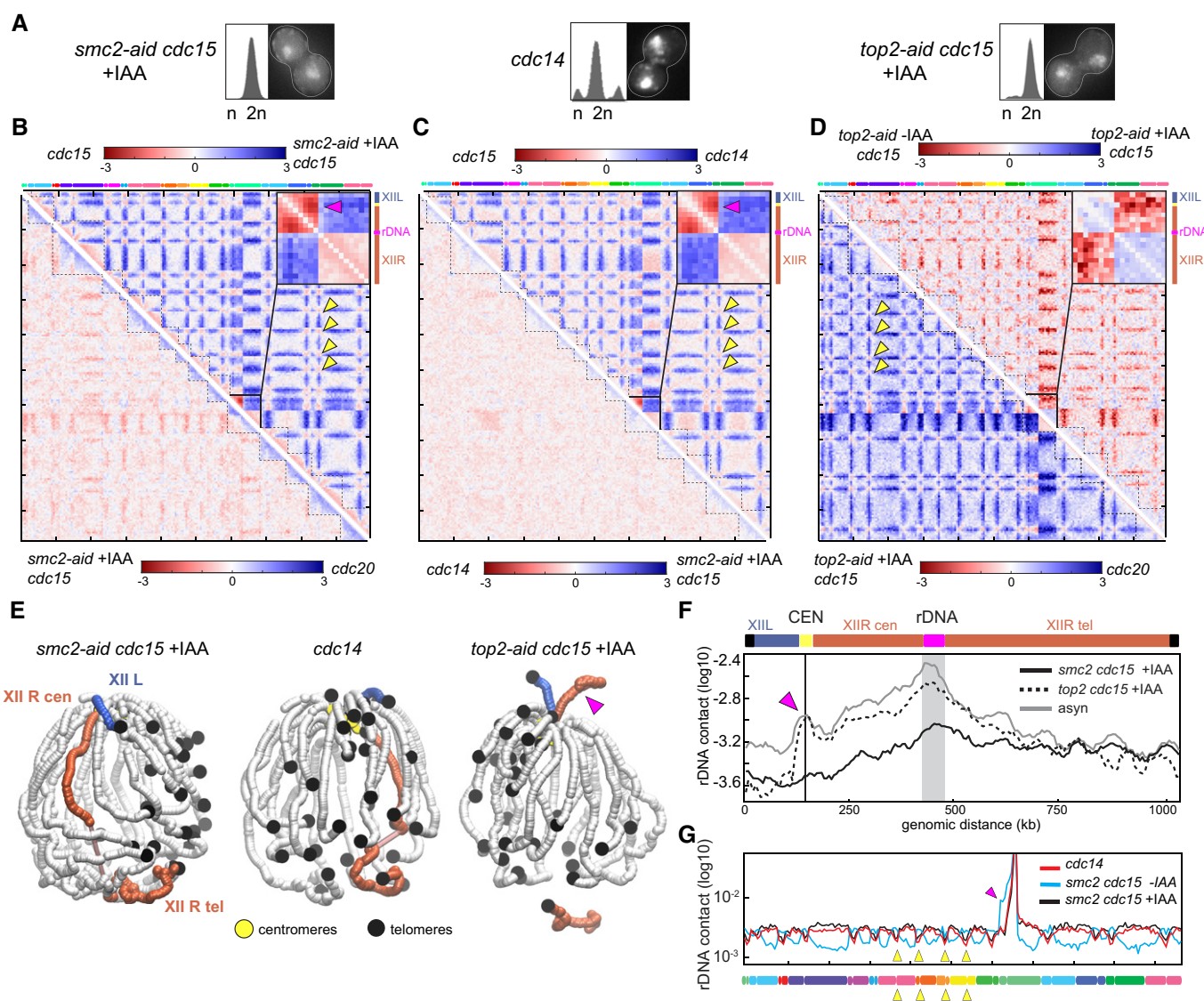

**Figure 5. The anaphase rDNA loop is condensin-dependent.**

A   FACS profiles and representative DAPI-stained cells of cells blocked in anaphase, in the absence of condensin (*smc2-aid cdc15* +IAA and *cdc14*) or topoisomerase 2 (*top2-aid cdc15* +IAA).

B–D   Log-ratio of contact maps. Yellow arrowheads: inter-centromere contacts. The pink arrowheads point at the right arm anaphase loop on chr12. Insets display magnification of the chr12 ratio map. (B) Ratio map between (bottom left) *cdc20* and *smc2-aid cdc15* cells and between (top right) cells blocked in anaphase with our without condensin depletion (*cdc15* and *smc2-aid cdc15* +IAA). (C) Ratio map between (bottom left) *cdc14* and *smc2-aid cdc15* +IAA cells and between (top right) *cdc14* and *cdc15* cells. (D) Ratio map between (bottom left) *top2-aid cdc15* +IAA and *cdc20* cells and (top right) *top2-aid cdc15* -IAA and *top2-aid cdc15* +IAA cells.

E   3D representations of the contact maps from *smc2-aid cdc15* +IAA-, *cdc14*-, and *top2-aid cdc15* +IAA-arrested cells. The right (XIIR) and left (XIIL) arms of chr12 are highlighted in red and blue, respectively. Pink arrowhead points at the right arm anaphase loop.

F   Distribution of intra-chromosomal contacts of a cen-proximal rDNA flanking region (highlighted in gray) with the rest of chr12 in *smc2 cdc15*, *top2 cdc15*, and asynchronous (asyn) cells. Pink arrowhead points at the right arm anaphase loop.

G   Normalized contact frequencies between the left rDNA flanking region (50 kb) and the rest of the genome in *cdc15 smc2-aid* (-IAA) and *cdc15 smc2-aid* (+IAA) cells. Yellow arrowheads point at a subset of centromeric positions. Pink arrowhead points at the right arm anaphase loop.

The organization of the genome was also compared in *cdc15*-arrested cells in the presence or absence of Top2, *top2-aid cdc15*-arrested (Fig 5D, upper right ratio; Fig 5E and F; Materials and Methods). Top2-depleted cells display a strong decrease in contacts between the telomere–proximal region of chr12R and the rest of the genome (including chr12L). The signal is consistent with the

essential role played by Top2 in rDNA segregation, showing that the non-segregated regions are isolated from the segregated chromosomal sets. The comparison between *top2 cdc15* and *cdc20* cells reveals an enrichment in contacts at centromeres and the persistence of the centromere–rDNA loop in the Top2 mutant (Fig 5D, bottom left ratio). These results indicate that the formation of these

## Discussion

This study consists of an experimental and analysis framework to systematically investigate and compare chromosome folding and organization at different stages of the cell cycle. We applied Hi-C to populations of cells synchronized at different points of the cycle, generating genome-wide, 5-kb-resolution contact maps which unveil their average 3D genome organization. The global influence of cohesin, condensin, and topoisomerase 2 has been investigated in the corresponding mutants, as well as the effects of the microtubule-depolymerizing drug nocodazole. Comparative approaches between contact maps provided a global view of the structural transitions between the different stages of the cycle, some expected, such as chromosome compaction during replication, and others that had not been described before, such as topological structures involving the rDNA cluster.

An overview of chromosome structural changes during the cell cycle can be summarized from centromere contacts, intra-/inter-chromosomal contact ratio, and short-/long-range contact ratio computed for each of the time points (Fig 6A).

Centromere clustering gradually increases during the cell cycle, through the establishment of sister-chromatid cohesion during replication, and through condensin-dependent clustering during anaphase (Fig 6A, upper panel). A potential consequence of this increased clustering in anaphase could be the generation of a stronger polymer brush, that is, the mechanical phenomenon that leads to the self-organization of a polymer tethered to a surface into stretched, non-intermingling structure (de Gennes, 1987). Interestingly, the strengthening of the polymer brush organization could consequently contribute to chromosome individualization during

condensin-dependent structures in anaphase is independent from the decatenation and/or the segregation of the rDNA cluster.

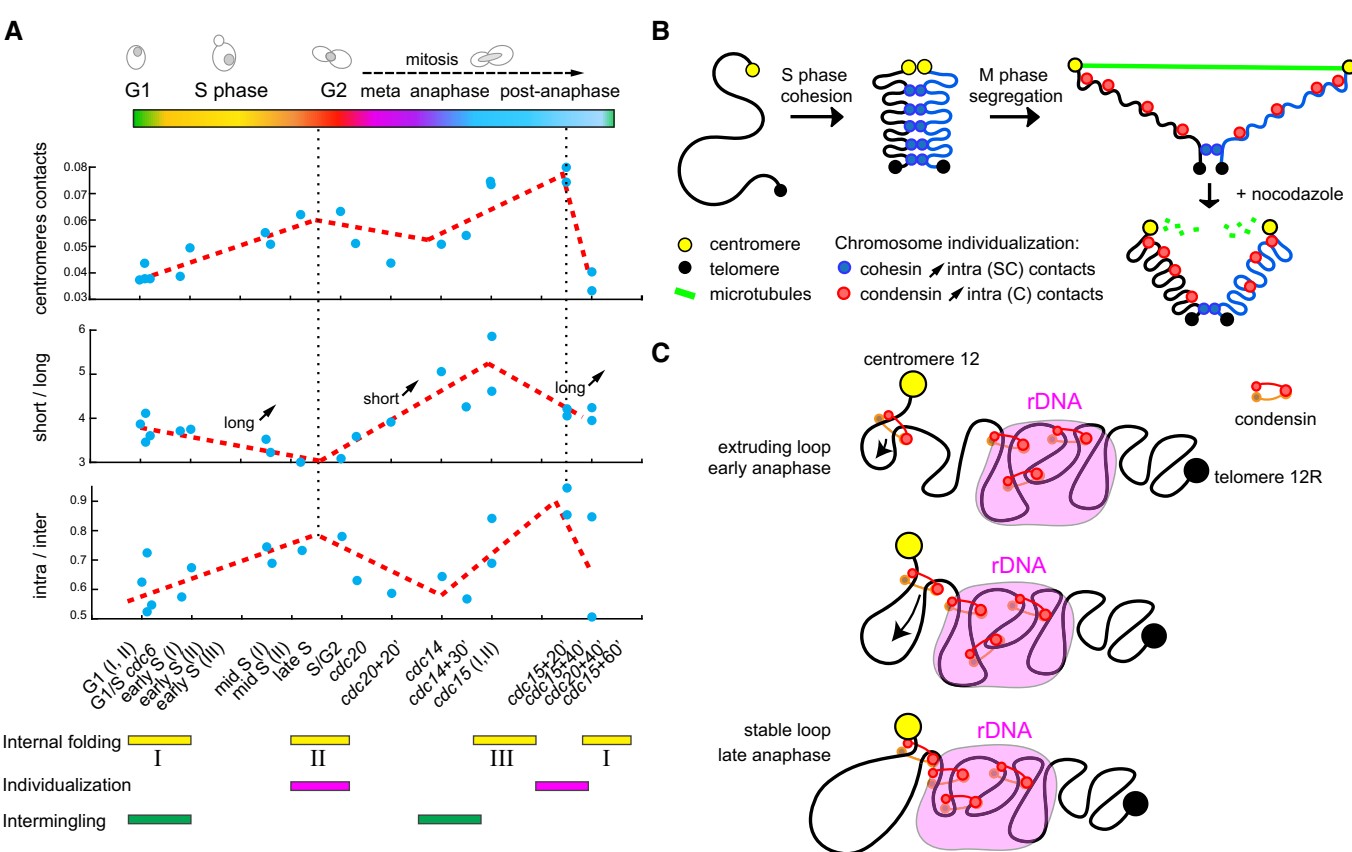

**Figure 6. 4D reorganization of the yeast genome.**

A Dynamics of centromere contacts (top panel), Short-/long-range contact ratio (middle panel) and intra-/inter-chromosomal contact ratio (bottom panel) for each of the 20 time points (blue dots; see bottom x-axis) during the cell cycle. The three folding states (I, II, and III; Fig EV4) identified in the analysis are indicated under the panels, as well as interpretation with respect to individualization status.

B Illustration of the three chromatin folding states characteristic of each of the cell cycle phases. Establishment of sister-chromatid (SC) cohesion during S phase increases intra-SC long-range contacts and leads to the individualization of the replicated chromosomes. Then during M phase, the two sisters are segregated and each chromatid (C) individualized thanks to the action combination of cohesin cleavage, condensin loading and spindle elongation. The chromosomes display a stretched internal structure, which relaxed upon destabilization of the spindle with nocodazole.

C Model of loop extrusion generating the condensin-dependent loop formation between the centromere and the rDNA cluster, two regions enriched in condensin deposition. A loop formed in between the centromere and the rDNA cluster may extend until it reaches these two discrete positions, and stall because of mechanic impediment blocking further extrusion.

anaphase. The intra-/inter-chromosomal contact variations reflect the successive phases of chromosome individualization and inter-mingling, with individualization taking place during replication (cohesin-dependent) and during anaphase (spindle-dependent; Fig 6A, bottom panel). The intra-/inter-chromosomal contact ratio correlates strongly with centromere clustering (c = 0.72, $p = 10^{-4}$), with both ratios peaking during anaphase exit.

Short-/long-range contact ratio recapitulates the three different internal folding (I, II, and III) states of chromosomes (G1, G2, and anaphase; Fig 6B, middle panel). These three states can be deter-mined based on a quantitative analysis of the significance of changes between $P(s)$ curves obtained using several replicates in different phases of the cycle (Fig 7). During replication, cohesins mediate the compaction of chromosomes from state I to II. The chromosomes are then stretched by the mitotic apparatus during anaphase (state III) before returning to state I in G1. The mechanical constraint imposed by the anaphase spindle appears responsible for the state III stretching, as a nocodazole treatment results in relax-ation of chromosomes, which switch back to state II. Imaging of the two sets of segregated chromosome during nocodazole treatment supports this spring relaxation effect, with the two masses being brought back together upon the depolymerization of microtubules. The nature of the mechanical constraints remains unknown, but it is tempting, in light of our observation of chr12 behavior (below), to propose a role for condensins in actively promoting this move-ment. In this scenario, condensins could favor the segregation of sister chromatids by pulling the chromosomes toward the centro-mere cluster. As a result, the loss of microtubule and tethering to the SPB may lead cohesin to actively pull back the segregated region together. We anticipate that whether condensins play an active role

in the segregation of chromosomes in addition to the pulling force imposed by the microtubule spindle will be thoroughly investigated in the years to come.

In addition, we also show that the two main regions of condensin deposition, *that is,* the centromeres and the rDNA locus, are bridged during anaphase through a condensin-dependent mechanism resulting in a loop-like structure on the right arm of chromosome 12. Whether this structure is systematically found in all cells, or only in a subset of the population, remains to be determined through single-cell imaging approaches such as FISH analysis. Although the precise mechanisms of formation remain unknown as well as its functional importance, we show that the setting up of the loop depends on condensin. Several mechanisms can be envisioned for the generation of this loop. One possibility is that starting from regions with a high condensin density, an active mechanism such as DNA extrusion through the action of conden-sins would pull the centromere and the rDNA cluster together (Fig 6C). Condensin depletion (leading to disruption of the loop) is associated with segregation defects. Overall, this structure therefore appears to *de facto* play a role in the segregation of the rDNA clus-ter, potentially through the application of a force that would drag the rDNA region to the centromere cluster before the completion of anaphase. A consequence of this model, would be that a similar loop extrusion mechanism could facilitate the segregation of other chromosomes as well. In this case, one or more loops could actively facilitate the segregation of large regions of chromosomes toward the tethered centromeres, down the telomeric regions. Chromosome 12, in this scenario, would appear as an exception with the large rDNA cluster generating a physical barrier in the middle of the right arm that is not present in other chromosomes. More experiments are nevertheless needed to investigate this proposed role. Yeast chromosome 12 could therefore prove a convenient model to study the action of loop extrusion mechanism (Alipour & Marko, 2012).

The importance of the rDNA loop remains to be further charac-terized as well as its similarity with loops found in other eukaryotic species. Overall, our exhaustive dataset opens new avenues for the comprehensive analysis of the 3D chromosome choreography during replication and segregation and brings to light new perspec-tives regarding these fundamental processes.

## Materials and Methods

### Media and culture conditions

All strains were grown in rich medium (YPD: 1% bacto peptone (Difco), 1% bacto yeast extract (Difco), and 2% glucose), except for YKL051 (*MET3-HA-CDC20*) that was grown in synthetic complete medium deprived of methionine (SC: 0.67% yeast nitrogen base without amino acids (Difco), supplemented with a mix of amino acids, uracil and adenine, and 2% glucose). Cells were grown at either 30°C or 23–25°C (the later temperature corresponding to the permissive temperature of the conditional thermosensitive muta-tions *cdc6-1, cdc14-3,* and *cdc15-2*; see below for details). Dataset corresponding to the quiescent state (G0) comes from already published data by Guidi *et al* (2015) and was obtained by carbon source exhaustion. All strains are described in Table EV1.

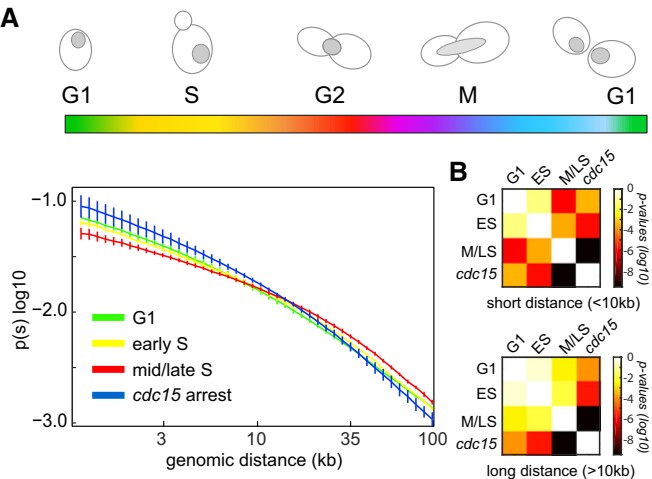

**Figure 7.  Variation in *P(s)* for different phase of the cell cycle.**

A  *P(s)* for four different time points along the cycle. Each curve represents the average between three replicates with error bars corresponding to the standard deviation.

B  To assess the statistical significance of the differences between short (resp. long)-range contacts between these three time points, we computed a *P*-value for each pairwise comparison between two time points using Wilcoxon signed-rank test. The two distributions to compared were built by aggregating all the data points below (resp. above) 10 kb for the three replicates for each time point.

## Elutriation (recovery of G1 cells)

To recover G1 daughter cells, the exponentially growing cultures were elutriated—a physical method of synchronization, used to separate cells according to their density and sedimentation velocity (see Appendix Supplementary Methods; Marbouty *et al*, 2014). The G1 daughter cells recovered through elutriation were suspended in fresh YPD at 30°C for 30 min, so they could recover from the elutriation procedure (i.e., stay in PBS). To minimize the potential variability introduced by the age heterogeneity of the bulk population, G1 daughter cells were used as starting point for all cell cycle synchrony and in combination with genetic and chemical synchronization methods (see below).

## Release into S phase

G1 elutriated cells were released into S phase to analyze genome conformation during this stage. $2 \times 10^9$ G1 cells—originating from the same elutriated fraction to minimize heterogeneity in replication initiation—were inoculated into 150 ml YPD at 25°C (to slow down replication fork progression). Upon release, the synchronized cultures were sampled every 5 min and the cells analyzed through FACS, revealing an approximate lag of 130 min before replication restart. Therefore, aliquots were cross-linked and processed into Hi-C libraries at 135, 140, 145, 150, 155, 160, and 165 min. The progression of each fraction throughout the S phase (from G1 to G2) was monitored with flow cytometry.

## Synchronization through thermosensitive mutations

Synchronizations using thermosensitive (ts) *cdc* strains (Hartwell *et al*, 1973) were all performed starting from elutriated G1 daughter cells growing in non-permissive temperature conditions designed to arrest the progression of the cycle at specific phases. See Appendix Supplementary Methods for details of synchronization procedures of strains YKL052 (*cdc14-3*), YKL053 (*cdc15-2*), and YKL054 (*cdc6-1*).

## Synchronization through chemical compounds

Chemical synchronization was also performed on elutriated G1 daughter cells.

Synchronization at the G2/M transition was achieved by restarting G1 cells (strain YKL050) in YPD at 30°C for 1 h, followed by the addition of nocodazole (Calbiochem; 15 µg/ml) and incubation for another 2 h at 30°C. Cells arrested in G2/M with nocodazole were either processed into Hi-C libraries, or washed and inoculated in fresh YPD medium at 30°C. The washing of nocodazole allowed G2/M synchronized cells to proceed into M phase (cells sampled after 20, 45, 60 and 90 min were processed into Hi-C libraries).

To investigate the constraints imposed by the spindle during anaphase, elutriated YKL053 cells were elutriated and the recovered G1 daughter cells processed and blocked into anaphase using the *cdc15-2* thermosensitive mutation. A sample of the population was then incubated with nocodazole (15 µg/ml) for 20 min. A sample was released at permissive temperature in the presence of nocodazole for 20 min. Finally, a sample was released at the permissive

temperature for 20 min before being incubated with nocodazole for 20 min.

For synchronization in metaphase, a system allowing induced depletion of *cdc20* was used (MET3-HA-CDC20; strain YKL051). Elutriated G1 daughter cells were restarted in YPD complemented with 50 µg/ml methionine for 5 h at 30°C. Cells arrested in metaphase were split into different aliquots. One sample was immediately processed into a Hi-C library, while two others were washed, suspended in SC medium without methionine, and processed into Hi-C after 20 and 40 min.

To investigate the influence of SMC on chromosome organization, strains carrying auxin-inducible degron (*aid*) versions of Scc1 (strain YKL055) and Smc2 (YKL056) proteins were processed into Hi-C libraries. The degradation of these proteins is induced when auxin (IAA) is added to the medium at a final concentration of 2 mM. Both asynchronous populations of strains YKL055 and YKL056 were elutriated in the absence of IAA. G1 daughter cells were incubated in YPD supplemented with IAA at 30°C. A sample of the YKL055 population (*scc1-aid*) was processed into a Hi-C library in late S/G2 (see Release into S phase). For the YKL056 population (*smc2-aid*), the cells were arrested in late anaphase using the *cdc15-2* mutation also present in the genome of this strain, before being processed into a Hi-C library.

To study the influence of topoisomerase II-mediated decatenation on chromosome organization, we used a strain (YKL057) in which *TOP2* gene is tagged by *aid* (*top2-aid*) and that also carries the *cdc15-2* mutation. An asynchronous exponentially growing culture of YKL057 cells was split into two fractions incubated for 3 h at the non-permissive temperature of 37°C in either the presence or absence of IAA (20 mM). The synchrony of each time point was monitored with flow cytometry and microscopy, and the cells were processed by Hi-C.

## Flow cytometry

About $5 \times 10^6$ cells were fixed in ethanol 70% and stored at 4°C overnight. Cells were then pelleted, washed, and incubated in sodium citrate 50 mM (pH 7.4) complemented with RNase A (10 mg/ml; Roche) for 2 h at 37°C. Next, Sytox green (2 µM in sodium citrate 50 mM; ThermoFisher) was added and cells incubated for 1 h at 4°C. Flow cytometry was performed on a MACS-Quant Analyzer (Miltenyi Biotec), and data were analyzed using FlowJo X 10.0.7 software (Tree Star).

## Microscopy

Fractions of cells fixed in ethanol 70% and stored at 4°C overnight were pelleted and washed three times for 5 min in 1× PBS. Cells were permeabilized by immersion in 0.2% Triton X-100 (Biosolve) for 5 min. To remove the Triton, cells were pelleted and washed three times in 1× PBS. The liquid was aspirated and cells were suspended in DAPI labeling solution (2 µg/ml in 1× PBS) for 10 min at room temperature. Before imaging acquisition, the labeling solution was aspirated and the cells were washed three times for 5 min in 1× PBS. Cells were imaged at 350 nm excitation wavelength with Nikon fluorescence microscope (Camera Andor Neo sCMOS, software Andor IQ2 2.7.1, LED Lumencor Spectra X).

## Hi-C libraries

Hi-C libraries were generated using the four-cutter enzyme DpnII through a protocol adapted from Belton *et al* (2012). The protocol is detailed in Appendix Supplementary Methods. The resulting libraries were used as template for the Illumina amplification by PE-PCR primers and paired-end-sequenced on the NextSeq 500 or HiSeq 2000 Illumina platform (2 × 75 or 2 × 150 bp kits; see Table EV2 for details).

## Generation and normalization of contact maps

Raw Hi-C data were processed as follows. PCR duplicates were removed using the 6 Ns present on each of the custom-made adapter and the 2 trimmed Ns. Paired-end reads were mapped independently using Bowtie 2.1.0 (mode: –very-sensitive –rdg 500,3 –rfg 500,3) against the *S. cerevisiae* reference genome (S288C). An iterative alignment, with an increasing truncation length of 20 bp, was used to maximize the yield of valid Hi-C reads (mapping quality > 30). Only uniquely mapped reads were retained. On the basis of their DpnII restriction fragment assignment and orientation, reads were classified as either valid Hi-C products or unwanted events to be filtered out (i.e., loops and non-digested fragments; for details, see Cournac *et al*, 2012, 2016). To generate contact matrices for all time points along the cycle, filtered Hi-C reads were binned into units of single restriction fragments, and then, successive fragments were assigned to fixed size bins of either 5 or 50 kb. Bins that exhibited a high contact frequency variance (< 1.5 Standard Deviation or 1.5–2 SD. from the mean) were filtered out for all maps to allow pairwise comparison of the data. On average, around 15 million of valid reads were used to build each contact map. To remove potential biases resulting from the uneven distribution of restriction sites and variation in GC content and mappability, the contact maps were normalized using the sequential component normalization (SCN) procedure (Cournac *et al*, 2012).

## Similarity between contact maps

To assess the similarity between normalized matrices, these were binned at 50 kb and quantile-normalized (Hicks & Irizarry, 2015). We then measured their similarity by computing the Euclidean distance between them. In order to visualize similarities between sets of matrices, we did a principal component analysis (PCA) of the pairwise distance matrix between samples.

## Contact probability within increasing genomic distance

Polymers display a decrease in contact probability, $P(s)$, as a function of the genomic distance, $s$. The degree of decay of $P(s)$ was often interpreted as informative of the polymer state. To compute the intra-chromosomal $P(s)$ plots, pair of reads aligned in intra-chromosomal positions were partitioned by chromosome arms. Reads oriented toward different directions or separated by < 1.5 kb were discarded to filter for self-circularizing events. For each chromosome, read pairs were log-binned in function of their genomic distance $s$ (in kb), according to the following formula:

$$bin = [\log 1.1(s)]$$

The $P(s)$ plot is the histogram computed on the sum of read pairs for each bin. This sum is weighted by the bin size $1.1^{(1+bin)}$ (because of the log-binning), as well as the difference between the length of the chromosome and the genomic distance $s$. The difference acts as a proxy for the number of possible events.

## 4C-like interaction plots

To obtain the 4C-like intra- and inter-chromosomal contact profiles for rDNA and centromeres, adjacent bins were indexed on the respective chromosomes. The resulting indexed and filtered matrices at either 5- or 50-kb bin were normalized using SCN (see Generation and normalization of contact maps). The profiles for the selected bins were plotted and compared using Matlab (no smoothing was applied).

## Computation of the replication profile from Hi-C data

The replication profile was computed from the raw 5-kb-binned contact maps. Firstly, G1 replicates were averaged and the sum of contact over each 5-kb bin was computed. The same computation was repeated for datasets obtained from cells released into S phase. To obtain the replication timing, we computed the ratio of these two signals and smoothed this ratio using a running-average window of six bins.

## 3D representation of contact maps

The 3D representations of the contact maps were generated using ShRec3D (Lesne *et al*, 2014) on the normalized contact maps, filtered for low-signal bins. First, the algorithm computes the distance matrix from the contact map, by assuming that distances are inversely proportional to the normalized contact counts. A shortest path algorithm is then used to insure that the distance matrix satisfies the triangular inequality. Finally, we used Sammon mapping to recover the optimal 3D coordinates from the distance matrix (Morlot *et al*, 2016). All the 3D structures presented here were rendered using VMD (Humphrey *et al*, 1996). Besides the cautiousness regarding the interpretation of 3D structure we mention in the main text, we also underline that the 3D structures are not used to compare datasets: All computational analyses are performed using the contact map data.

## Comparison of centromeres, intra-/inter-, and short-/long-range contacts between datasets

To compare contacts between centromeric regions, the sum of normalized inter-chromosomal contacts between 100-kb regions centered on centromeres was computed and divided by the total number of normalized inter-chromosomal contacts between all chromosomes. To compare short- versus long-range contacts, a ratio of intra contacts was computed as follows. The number of intra contacts involving fragments positioned < 30 kb apart was divided by the number of intra contacts involving fragments positioned more than 30 kb apart, for all chromosomes. For intra- versus inter-chromosomal contacts, the total number of normalized

intra-chromosomal contacts was divided by the sum of normalized inter-chromosomal contacts.

### Quantification of variability between replicates

To assess for the contribution of experimental variability to the variations in contacts between different conditions, we proceeded as follows. Density histograms displaying the distribution of the log2 contact ratio of all elements of Hi-C matrices (50-kb bins) between pairs of biological and experimental replicates (3×G1, 2×G2, 3×M) were computed and compared to similar histograms computed from pairs of Hi-C matrix obtained in different experimental conditions (see Appendix Fig S2).

An estimation of the replicate variability at the centromeres was obtained by plotting the boxplots representing the distribution of the log2 contact ratios between pairs of biological and experimental replicates only of the bins encompassing the centromeres (50-kb bins; see mask; Appendix Figs S3 and S4). The same computation was performed on pairs of matrices obtained in different conditions to estimate the statistical significance of the variations. All replicates were taken into account. *P*-values were obtained by the Wilcoxon signed-rank test.

**Expanded View** for this article is available online.

### Acknowledgements

We thank Martial Marbouty and Axel Cournac for contributions to the early stage of this project. Sample description and raw contact maps are accessible on the GEO database through the following accession number: GSE90902. Raw sequences are accessible on SRA database through the following accession number: SRP094582. We are also grateful to Armelle Lengronne, Stephane Marcand, Philippe Pasero, Etienne Schwob, Emmanuelle Fabre, Nancy Kleckner, and Angela Taddei for sharing strains and for discussions. Vittore Scolari and Heloise Muller were partly supported by Pasteur-Roux-Cantarini postdoctoral fellowships. This research was supported by funding to R.K. from the European Research Council under the 7th Framework Program (FP7/2007-2013, ERC grant agreement 260822), from Agence Nationale pour la Recherche (MeioRec ANR-13-BSV6-0012-02), and from ERASynBio and Agence Nationale pour la Recherche (IESY ANR-14-SYNB-0001-03).

### Author contributions

LL-S and RK designed research. LL-S performed the experiments, with contributions from GM, AT, and HM. TMG generated the smc2-aid mutant. VFS and JM analyzed the data, with contributions from LL-S. LL-S, JM, and RK interpreted the data and wrote the manuscript.

### Conflict of interest

The authors declare that they have no conflict of interest.

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
