## [Review Process File · The EMBO Journal]

Manuscript EMBO-2017-97342

Cohesins and condensins orchestrate the 4D dynamics of yeast chromosomes during the cell cycle

Luciana Lazar-Stefanita Vittore F. Scolari, Guillaume Mercy, Héloïse Muller, Thomas M Guérin, Agnès Thierry, Julien Mozziconacci and Romain Koszul

*Corresponding authors: Julien Mozziconacci, Sorbonne Universités
Romain Koszul, Institut Pasteur*

Review timeline:	Submission date:	21 May 2017
	Editorial Decision:	01 June 2017
	Revision received:	14 June 2017
	Editorial Decision:	26 June 2017
	Revision received:	29 June 2017
	Accepted:	04 July 2017

Editor: Anne Nielsen

Transaction Report:

(Note: Please note that the manuscript was previously reviewed at another journal and the reports were taken into account in the decision making process at The EMBO Journal. Since the original reviews are not subject to EMBO's transparent review process policy, the reports and author response cannot be published.)

1st Editorial Decision	01 June 2017
--------------

Thank you for submitting your manuscript to The EMBO Journal, along with the referee reports and your response from the previous round of peer review at another journal. We have now consulted with an arbitrating advisor who had access to your manuscript, the referee reports, and your response - and this person supports publication here following minor revision as outlined below.

As you will see, our advisor highlights the quality and quantity of your data but at the same time points out that you should discuss the validity in depicting 3D maps of chromome organization/interactions based on data averaging numerous cells. In addition, the advisor finds that cross-validation using a separate technical approach such as FISH would significantly strengthen the conclusiveness of the HiC-derived interactions. I realise that doing this at a broader scale would be outside the scope of the current study but in case you have data available for a few contact points (as a proof of principle) I would suggest that you include it in the final version of the manuscript. Finally, you will see that our advisor would like to see an extended discussion section.

Based on the overall positive recommendation from our advisor I would invite you to submit a revised version of the manuscript in which you address the points outlined above.

Thank you again for giving us the chance to consider your manuscript for The EMBO Journal, I look forward to your revision.

 REFEREE REPORTS

Referee #1:

Lazar-Stefanita et al. use hi-C methods to describe the changes in higher order genome organization through the yeast cell cycle. Not surprisingly, they find considerable re-organization of genome interactions at various stages of the cell cycle, including cell division. Using mutants they provide evidence for a role of cohesins in S-phase re-organization and condensin in anaphase re-organization.

The study is a Hi-C tour the force and is of good technical quality. The results are somewhat descriptive and largely expected, but will be of interested to the specialists in the field.

Specific points to be addressed:

1. Although widely used in the community, the use of the 3D representations based on 2D maps is misleading. The datasets represent ensemble information from millions of cells and the 3D representations give the impression that the population is homogenous, i.e. all cells in the population have the same 3D organization. We know from single cell hi-C that this is not the case. The authors should 1) provide some indication as to how many interactions are on average detected per analyzed cell, 2) should, if possible, provide a sense of heterogeneity in the population and 3) should, at least, clearly indicate in the text that what they observe are averaged ensemble representations.
2. The study lacks orthogonal validation of many of the conclusions. Ideally the authors should perform FISH experiments to test the validity of some of the key conclusions such as the formation of the condensin-dependent chromosome 12 rDNA loop and the effect of condensin on anaphase organization by localizing specific interaction sites and demonstrating altered spatial relationships.
3. As a minor point, the Conclusion session seems to have been written in haste. It is very superficial and should be re-written with a little more care.

1st Revision - authors' response

14 June 2017

Response to referees:

Referee #1:

Lazar-Stefanita et al. use hi-C methods to describe the changes in higher order genome organization through the yeast cell cycle. Not surprisingly, they find considerable re-organization of genome interactions at various stages of the cell cycle, including cell division. Using mutants they provide evidence for a role of cohesins in S-phase re-organization and condensin in anaphase re-organization.

The study is a Hi-C tour the force and is of good technical quality. The results are somewhat descriptive and largely expected, but will be of interested to the specialists in the field.

Specific points to be addressed:

1. Although widely used in the community, the use of the 3D representations based on 2D maps is misleading. The datasets represent ensemble information from millions of cells and the 3D representations give the impression that the population is homogenous, i.e. all cells in the population have the same 3D organization. We know from single cell hi-C that this is not the case. The authors should 1) provide some indication as to how many interactions are on average detected per analyzed cell, 2) should, if possible, provide a sense of heterogeneity in the population and 3) should, at least, clearly indicate in the text that what they observe are averaged ensemble representations.

The methods was actually very explicit about the fact that the Hi-C 2D and 3D maps are average representations (as highlighted below).

3D Representation of contact maps. The 3D representations of the contact maps were generated using ShRec3D (Lesne et al, 2014) on the normalized contact maps, filtered for low signal bins.

First, the algorithm computes the distance matrix from the contact map, by assuming that distances are inversely proportional to the normalized contact counts. A shortest path algorithm is then used to insure that the distance matrix satisfies the triangular inequality. Finally we use Sammon mapping to recover the optimal 3D coordinates from the distance matrix (Morlot et al, 2016). All the 3D structures presented here were rendered using VMD (Humphrey et al, 1996). These 3D structures are average representations and therefore do not represent the exact structure found in an individual cell. It must also be underlined that they are not polymer models and cannot be interpreted as such. They have to be interpreted in light of the contact frequencies over a population of cells. For instance, on these 3D representations the telomeres loosely cluster together. In a single nucleus, telomeres would rather form smaller groups scattered all around the nuclear membrane. Since in different cells these group gather different partners, they are regrouped together in the average structure that reflects the population average of contacts. We also underline that the 3D structures are not used to compare datasets: all computational analysis are performed using the contact map data.

We have moved part of these explanations in the main text, to be even more explicit. As for the number of contacts per cell, our sequencing depth is far from being exhaustive (we have only a few % of duplicates in our libraries). Therefore, the ratio corresponding to 20M reads (on average) generated over a population of ~1bn cells is not informative (and is not a metric used in any Hi-C, ChIP or RNA-seq paper).

Here is the new main text paragraph explaining how to interpret the 3D structures:

These 2D maps were translated into 3D representations to visualize the main folding features (Lesne et al, 2014) (e.g. centromeres and telomeres clustering in G1, Fig 1B; Fig EV1). These 3D structures are average representations of the contact frequencies quantified over a population of cells, and therefore do not represent the exact structure found in individual cells. For instance, on these 3D representations all the telomeres loosely cluster together. In a single nucleus, telomeres rather form small groups scattered all around the nuclear membrane (Taddei & Gasser, 2012). Since in different cells the composition of these clusters differ, all telomeres end up being regrouped together in the average 3D structure that reflects the population average of contacts. In addition, they are not polymer models, and cannot be interpreted as such. Nevertheless, these representations conveniently highlight important structural features not readily apparent in the 2D maps.

2. The study lacks orthogonal validation of many of the conclusions. Ideally the authors should perform FISH experiments to test the validity of some of the key conclusions such as the formation of the condensin-dependent chromosome 12 rDNA loop and the effect of condensin on anaphase organization by localizing specific interaction sites and demonstrating altered spatial relationships. We agree that backing several of our observations with FISH analysis would be ideal. However, we think this is a future work that needs to be also tacked on the basis of the present study. We have added sentences in the discussion that clearly state that the interpretation we propose for some of our data will require further investigation, notably through single cell approaches, to be fully validated. In addition, we also show that the two main regions of condensin deposition, i.e. the centromeres and the rDNA locus, are bridged during anaphase through a condensin-dependent mechanism resulting in a loop-like structure on the right arm of chromosome 12. Whether this structure is systematically found in all cells, or only in a subset of the population, remains to be determined through single cell imaging approaches such as FISH analysis. Although the precise mechanisms of formation remains unknown as well as its functional importance, we show that the setting up of the loop depends on condensin.

3. As a minor point, the Conclusion session seems to have been written in haste. It is very superficial and should be re-written with a little more care.

We have converted the conclusion section into a discussion.

2nd Editorial Decision

26 June 2017

Thank you for submitting a revised version of your manuscript, I am pleased to inform you that it is now in principle ready for acceptance here. However, before we can go ahead and transfer your manuscript files for production there are still a few formatting issues that need to be resolved.

2nd Revision - authors' response

29 June 2017

Authors made requested editorial changes.

Corresponding Author Name: KOSZUL

Manuscript Number: EMBOJ-2017-97342